# Polymer/Layered Clay/Polyurethane Nanocomposites: P3HB Hybrid Nanobiocomposites—Preparation and Properties Evaluation

**DOI:** 10.3390/nano13020225

**Published:** 2023-01-04

**Authors:** Anita Białkowska, Beata Krzykowska, Iwona Zarzyka, Mohamed Bakar, Vladimir Sedlařík, Miroslava Kovářová, Anna Czerniecka-Kubicka

**Affiliations:** 1Faculty of Chemical Engineering and Commodity Science, University of Technology and Humanities, Chrobrego 27, 26-600 Radom, Poland; 2Faculty of Chemistry, Rzeszów University of Technology, Powstańców Warszawy 6, 35-959 Rzeszów, Poland; 3Centre of Polymer Systems, University Institute, Tomas Bata University in Zlin, Tr. T. Bati 5678, 76001 Zlin, Czech Republic; 4Department of Experimental and Clinical Pharmacology, Medical College of Rzeszow University, The University of Rzeszow, 35-310 Rzeszow, Poland

**Keywords:** nanobiocomposites, polyalcanoates, nanoclay, modified montmorillonite, properties modification, aliphatic polyurethane

## Abstract

This paper presents an attempt to improve the properties of poly(3-hydroxybutyrate) (P3HB) using linear aliphatic polyurethane (PU400) and organomodified montmorillonite (MMT)—(Cloisite^®^30B). The nanostructure of hybrid nanobiocomposites produced by extrusion was analyzed by X-ray diffraction and transmission electron microscopy, and the morphology was analyzed by scanning electron microscopy. In addition, selected mechanical properties and thermal properties were studied by thermogravimetric analysis, TGA, and differential scanning calorimetry, DSC. The interactions of the composite ingredients were indicated by FT IR spectroscopy. The effect of the amount of nanofiller on the properties of prepared hybrid nanobiocomposites was noted. Moreover, the non-equilibrium and equilibrium thermal parameters of nanobiocomposites were established based on their thermal history. Based on equilibrium parameters (i.e., the heat of fusion for the fully crystalline materials and the change in the heat capacity at the glass transition temperature for the fully amorphous nanobiocomposites), the degree of crystallinity and the mobile and rigid amorphous fractions were estimated. The addition of Cloisite^®^30B and aliphatic polyurethane to the P3HB matrix caused a decrease in the degree of crystallinity in reference to the unfilled P3HB. Simultaneously, an increase in the amorphous phase contents was noted. A rigid amorphous fraction was also denoted. Thermogravimetric analysis of the nanocomposites was also carried out and showed that the thermal stability of all nanocomposites was higher than that of the unfilled P3HB. An additional 1% mass of nanofiller increased the degradation temperature of the nanocomposites by about 30 °C in reference to the unfilled P3HB. Moreover, it was found that obtained hybrid nanobiocomposites containing 10 wt.% of aliphatic polyurethane (PU400) and the smallest amount of nanofiller (1 wt.% of Cloisite^®^30B) showed the best mechanical properties. We observed a desirable decrease in hardness of 15%, an increase in the relative strain at break of 60% and in the impact strength of 15% of the newly prepared nanobiocomposites with respect to the unfiled P3HB. The produced hybrid nanobiocomposites combined the best features induced by the plasticizing effect of polyurethane and the formation of P3HB–montmorillonite–polyurethane (P3HB-PU-MMT) adducts, which resulted in the improvement of the thermal and mechanical properties.

## 1. Introduction

With the growing awareness in society about environmental care, the use of biodegradable plastics is now the best solution to the environmental risks posed by conventional and commonly used plastics. Biodegradable plastics are environmentally friendly, and their properties can be compared with those of conventional polymers. Many applications have been, including in food packaging and in biomedicine [1,2,3,4,5,6,7,8,9,10,11,12,13,14,15,16,17,18]. The only 100% biodegradable polymers are polyhydroxyalkanoates (PHAs) [19,20]. One of the more important PHAs is poly(3-hydroxybutyrate), known as P3HB (I) Figure 1, which shows a much longer degradation time compared to polymers from the poly(α-hydroxyalcanoates) group, such as polylactide or poly(lactic-co-glycolic acid) [21].

Poly(3-hydroxybutyrate) is produced as an energy carrier from various bacteria [22,23,24], which build polymer chains that are perfectly linear and isotactic, allowing for unique properties and a high degree of crystallinity. In this form, P3HB is biobased, biodegradable and biocompatible [11,25,26]. However, this material shows some disadvantages (high stiffness, brittleness, and low thermal stability above the melting point) when compared to other polymers [27,28,29]. For these reasons, and in order to increase its range of applications, it has undergone multiple modifications [30]. In most cases, the production of composites based on the P3HB matrix leads to the much-desired separation of its melting point from its degradation temperature which characterizes their better thermal and mechanical properties than the unfiled P3HB.

Polymer nanocomposites are currently an important direction in industrial development. The addition of even a small amount of nanofillers to a polymer leads to improvement in their mechanical, thermal, barrier, flammability, ductility, stiffness, and electrical and thermal conductivity properties, without affecting their processability [30,31,32,33]. Different studies have confirmed that hybrid biopolymer nanocomposites have shown even better properties than conventional composites, including mechanical properties [34,35].

The present work investigated the effect of aliphatic linear polyurethane and organomodified montmorillonite (Cloisite^®^30B) on the performance properties and structure of P3HB biopolymer. It was expected that P3HB-based hybrid nanobiocomposites would exhibit improved properties due to two different reinforcing mechanisms of the used modifiers.

## 2. Materials and Methods

### 2.1. Materials

P3HB was supplied by Biomer (Krailling, Germany); its weight average molecular mass was M_w_ = 443,900 g∙mol^−1^ and its dispersity was (M_w_∙M_n_^−1^) = 5.72; the P3HB melt flow index was 0.11 g∙(10 min)^−1^ (180 °C at 2.16 kg). Organically modified montmorillonite Cloisite^®^30BB was supplied by Southern Clay Products Inc. (Gonzales, LA, USA). Cloisite^®^30BB is a natural montmorillonite modified with methylbis(2-hydroxyethyl)tallowalkylammonium cations.

Hexamethylene 1,6-diisocyanate (HDI) and dibutyltin dilaurate (DBTL) were purchased from Sigma-Aldrich (Saint Louis, MO, USA), and polyethylene glycol, with a molecular mass of 400 g/mol (PEG400), from Merck (Darmstadt, Germany).

#### Linear Polyurethane Synthesis

The molecular weight of the PU400 determined by size exclusion chromatography (SEC) in tetrahydrofuran (THF) was M_w_ = 13.430 g/mol and the dispersity was M_w_/M_n_ = 2.09. Linear polyurethane was synthesized by the procedure described in [36].



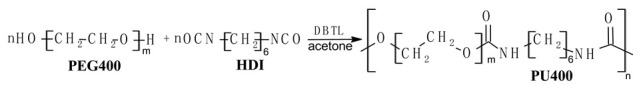



where m is 8.7 and n is 8.6 on average.

### 2.2. Methods

#### 2.2.1. Nanobiocomposite Preparation

To homogenize the components, P3HB was introduced into a Stephan-type mixer, followed by an appropriate amount of liquid PU400 and an appropriate amount of Cloisite^®^30B. The mixture was then stirred for about 20 min at room temperature for homogenization. Table 1 shows the compositions of the prepared mixtures.

The homogeneous mixture was then introduced into the hopper of a co-rotating twin-screw extruder with a working diameter of D = 25 mm and L/D = 33, operating at 300–450 rpm. The different zones of the extruder were maintained at the following temperatures during the extrusion of the composite: hopper: 22–36 °C, II zone: 120–125 °C, III zone: 135–145 °C, IV zone: 134–136 °C, V zone: 134–135 °C, VI zone: 145–148 °C, VII zone: 145–148 °C, VIII zone: 148–163 °C, IX zone: 134–150 °C, and head: 158 °C to 170 °C.

During the extrusion, the volatile parts were evacuated by degassing, and the temperatures of the head of the extruder and the heating zones of the plasticizing system of the extruder were kept constant. The molten composite was cooled in a cooling bath, pelletized, and dried at 60 °C for 2 h.

The unfilled P3HB and a mixture of P3HB containing 10 wt.% of PU400 (C-10-0) were also extruded in order to compare the properties of the prepared nanobiocomposites with a reference material.

#### 2.2.2. Analytical Methods

The hydroxyl number (HN) of polyethylene glycol and the obtained polyurethane were determined according to the standard (PN-87/C-89085/13). For this purpose, 1 g of the tested compound was placed in a conical flask and the acetylating mixture, i.e., a solution of acetic anhydride in DMF, was added. The flask was equipped with a reflux condenser, and the solution was heated in a boiling water bath for one hour. After heating was completed, toluene was poured into the flask and stirred vigorously. Then, it was titrated with 1.5 M potassium hydroxide solution in the presence of phenolphthalein until a persistent pink color appeared. The hydroxyl number value was calculated from the formula:(1)$$HN=\frac{56.1\cdot \left(V_{1}-V_{2}\right)\cdot C}{m}$$
where HN—hydroxyl number, (mg KOH/g), C—molar concentration of the KOH solution, (mol/dm^3^), V_1_, V_2_—volumes of the KOH solution used for the titration of the blank sample and the analyzed sample, respectively, (cm^3^), m—mass of the sample, (g).

The content of the isocyanate groups (ICN) in the synthesized polyurethane was determined according to the standard (PN-EN 1242:2006). A 0.5 g sample was placed in a conical flask, dissolved in acetone, and mixed with 0.2 M acetone ammonia solution. The flask was stoppered, the solution was stirred, and then left standing for 10 min at room temperature. The excess ammonia was titrated with 0.1 M HCl_aq_ in the presence of methyl red until a bright red color persisted. A blank test was performed in parallel. The contents of the isocyanate groups (ICN) in % by weight were calculated using the equation:(2)$$LIC=0.0042\cdot \frac{\left(V_{1}-V_{2}\right)}{m}\cdot 100\%$$
where V_1_ and V_2_—volumes of 0.1 M HCl_aq_ used for the titration of the blank and the sample, respectively, (cm^3^); m—mass of the sample, (g); 0.0042—the number of isocyanate groups equivalent to 1 cm^3^ of a 0.1 M HCl solution, (g).

#### 2.2.3. Instrumental Methods

The X-ray diffraction technique was used to characterize the structure of biocomposites at the nanometric scale. The measurements were carried out at room temperature with the use of a Bruker SAXS Nanostar-U X-ray diffractometer. The spectra of the samples were studied in transmission mode to verify the effect of nanoparticle orientation. A small-angle diffractometer was coupled to a filtered CuKa radiation source (1.54 Å) in a sealed tube, operating at 50 kV and 30 mA. The 2D detector (Vantec2000) was used to scan the entire sample surface; the spot beam was approx. 500 μm. The scanning range was determined by changing the distance of the sample from the detector. The resolution of the detector and the angular range allowed for measurement at the level of 2048 × 2048 pixels. The measurements were carried out from 1 to 28° over a period of 2 h.

The nanostructures of the nanobiocomposites were tested by transmission electron microscopy (TEM). The nanoparticle distribution was imaged using a Tecnai TF20 X-TWIN (200 kV)—a high-resolution transmission electron microscope equipped with a field emission gun (FEG) (Thermo Scientific, Waltham, MA, USA). TEM samples were prepared with the use of a microtome.

The study of the composite morphologies was carried out using a JEOL JSM-6490 LV scanning electron microscope with an accelerating voltage of 20 kV and secondary electron detection (SEI). The samples to be tested were subjected to freezing in liquid nitrogen and then broken using a blade hammer. The surfaces of the broken samples were then sputtered with a thin layer of gold with a thickness of about 10 nm using a JEOL JFC-1300 sputtering machine. Finally, the sputtered samples were placed in an electron microscope chamber, and the analysis of the material surface was conducted within several micro-areas.

The infrared (FT IR) spectra of the P3HB, polyurethane, and their composites were recorded on an ALPHA FT-IR Bruker instrument in the ATR mode or in KBr pellets.

Thermogravimetric analysis (TGA) of the nanobiocomposites was conducted using a Metler Toledo TGA/DSC 3+ thermogravimetric analyzer. The composite samples were heated at a rate of 5 °C/min in the temperature range of +25 to +600 °C in an inert gas (nitrogen) atmosphere. The temperature of the onset of decomposition (T_on_), the temperature corresponding to 50% weight loss (T_50%_), the temperature of the maximum decomposition rate (T_max_), and the total weight loss of the samples at 600 °C were determined.

All experiments by calorimetry in the temperature range from −90 °C to 195 °C were performed using a Q1000TM differential scanning calorimeter (DSC) from TA Instruments, Inc. (New Castle, DE, USA). This calorimeter is the heat-flux type and is equipped with a mechanical refrigerator to control the heating and cooling of samples. The measurements were carried out in a nitrogen atmosphere with a constant flow rate of around 50 mL/min. The mass of samples used for measurements by DSC was about 10 mg. The experimental heat flow was measured at a heating rate of 10 °C/min after previously cooling at a rate of 10 °C/min by the standard DSC. The temperature and heat flow rate calibration in the DSC apparatus was performed using the parameters of the melting indium, the onset melting temperature (T_m_(onset) = 156.6 °C), and the heat of fusion (ΔH_f_ = 28.45 J/g (3.281 kJ/mol)). In order to obtain accurate results, the heat capacity was calibrated by sapphire [37]. The heat flow data were collected from the second heating run after the controlled cooling.

The following mechanical tests were conducted: tensile strength, impact strength, and Shore hardness. The samples for mechanical testing were produced by injection molding technology on the DrBoy 50 injection molding machine equipped with the Priamus production supervision and monitoring system. All samples were produced with the same adjustable injection parameters, which made it possible to assess changes in the parameters of the material state in the injection mold. An injection mold temperature of 25 °C was used for the P3HB, and 30 °C for the nanobiocomposites, and the process temperature was in the range of 140–168 °C.

The tensile tests of the nanocomposites were carried out at room temperature using a Zwick Z030 testing machine equipped with an automatic multiextensometer, and with a deformation speed of 50 mm/min. Young’s modulus was determined at a rate of 1 mm/min. The tests were conducted for 10 samples in a given series in accordance with the standard (PN-EN ISO 527 (1998)).

The impact strength of the nanobiocomposite samples was performed using an Instron brand impact hammer. The impact test was carried out for bilaterally notched samples according to method A of the standard (PN-EN ISO 179-1:2010).

The hardness of the obtained nanobiocomposite samples was determined by the Shore method using Bareiss Shore hardness testers: HP-A and HP-D. The measurement consisted of measuring the sample resistance of the tested composite while immersing an indenter needle with a specific shape and dimension in accordance with the standard (PN-EN ISO 868 and ASTM D2240).

## 3. Results

### 3.1. Hybrid Nanobiocomposite Preparation

Hybrid nanobiocomposites based on P3HB with linear aliphatic polyurethane and modified montmorillonite (MMT) (Cloisite^®^30B) nanoadditive were produced using a co-rotating twin-screw extruder. Polyurethane obtained in the reaction of 1,6-hexamethylene diisocyanate and polyethylene glycol with a molar mass of 400 g/mol was used in the amount of 10 wt.%, and Cloisite^®^30B in the amounts of 1, 2, and 3 wt.%, (C-10-1, C-10-2, C-10-3). In order to obtain a reference material to compare the properties with the obtained nanobiocomposites, unfilled P3HB and a mixture of P3HB with 10 wt.% PU400 (C-10-0) were used.

### 3.2. Structure Analysis by X-ray Diffraction

To characterize the nanoclay structure of the prepared nanobiocomposites, the X-ray diffraction (XRD) technique was used in the range of 1° to 30°. For comparison, reference diffractograms of unfilled P3HB and Cloisite^®^30B are shown in Figure 1. The unfilled P3HB showed only low-intensity background scattering below 12°, while Cloisite^®^30B was characterized by a diffraction maximum at an angle of 2θ = 5.00°, which means that the plates were separated by a distance d_001_ of 1.77 nm. The obtained nanobiocomposites containing 1 wt.% of nanoclay (C-10-1) did not show a diffraction maximum below the value of 2θ < 5°. This means that the P3HB chains penetrated between the layers of the nanoclay platelets and there was not only separation, but complete delamination. On the other hand, the introduction of 2 and 3 wt.% of Cloisite^®^30B resulted in a significant broadening of the null peak. In addition, a low-intensity peak was observed with a diffraction maximum at 4.64°, corresponding to the second reflection (d_002_). Its presence suggests the possible small share of the partially intercalated structure of the nanobiocomposites.

Moreover, the diffractograms of the prepared nanobiocomposites indicate an increase in the amorphous degree of the semicrystalline matrix P3HB. This is evidenced by the shape change of peaks in the range of 2Θ from 19 to 27°.

### 3.3. Nanostructure Analyzed by TEM

The nanobiocomposites were observed by the TEM method to identify the nanostructure features. The structure analysis of the resulting nanobiocomposites by TEM confirmed the conclusions resulting from the XRD analysis. The TEM micrographs of the nanobiocomposites are shown in Figure 2. Additionally, the TEM micrograph of the unfilled P3HB was also added for comparison. The TEM images of all nanobiocomposites showed a high degree of dispersity of organically modified montmorillonite, i.e., Cloisite^®^30B. P3HB chains penetrated the silicate gallery, with complete delamination (exfoliation) on single silicate nanoplatelets. However, the influence of the nanofiller concentration was not observed at the level of exfoliation. To emphasize this state, the micrograph with less magnification (200 nm) was shown for the nanobiocomposite with the highest concentration (3 wt.%) of Cloisite^®^30B (C-10-3).

### 3.4. Morphology Analysis of the Obtained Nanobiocomposites

Figure 3 shows the SEM micrographs of the fractured surfaces of the polyester (P3HB) samples and biocomposites containing 10% polyurethane prepared from polyethylene glycol (PU400) and various amounts of nanofillers. The images shown were obtained by scanning the samples at the breaking point due to the applied force.

The fractured surface of the so-called null unmodified polyester (P3HB), shown in Figure 3, is slightly wavy and glassy, indicating the presence of a regular crack propagation path, and areas of brittle breakthrough are arranged unidirectionally.

The next micrograph shows the surface of the blend loaded with 10 wt.% of polyurethane based on PEG400. As can be seen, the introduction of polyurethane (PU400) as the only modifier caused an apparent disruption of the continuity of the P3HB matrix structure. The presence of crystalline domains is noticeable. The rough areas visible in the image, which are arranged in different directions, suggest the interaction of the biopolymer matrix with the PU400, resulting in the separation of the interacting P3HB chains and the formation of the mentioned domains.

The following micrographs in Figure 3 show the structures of the hybrid nanobiocomposites loaded with 10 wt.% of PU400 and different amounts of nanofiller (1%, 2%, or 3 wt.% of Cloisite^®^30B).

The polyester matrix modified with 10 wt.% of PU400 and the smallest amount of Cloisite^®^30B (1 wt.%) had a noteworthy completely different structure from the other samples. Distinct crystalline domains arranged in different directions were much larger in size than in the other accompanying photos. In addition, these domains were separated by larger distances than in the other samples, suggesting a smaller interaction between them and a flexibilizing effect of the P3HB.

### 3.5. Mechanical Properties of Nanobiocomposites

Figure 4a shows that the tensile strength (TS) of the obtained nanobiocomposites decreased with increasing amounts of incorporated nanofillers. Moreover, the addition of PU400 caused a 13.8% decrease in the TS, while the addition of Cloisite^®^30B caused a further decrease in the TS. The greater the addition of nanofiller, the greater the decrease in stress, and it reached about 17.5% for the nanobiocomposite containing 1 wt.% of Cloisite^®^30B (C-10-1) and 36.0% for the nanobiocomposite with 3 wt.% of Cloisite^®^30B (C-10-3). Figure 4b shows the strain at break as a function of the sample composition. A slight increase in the elongation of 11.4% can be seen due to the addition of PU400. The value of strain at break of the nanobiocomposite C-10-1 increased significantly, by about 60% compared to the unmodified sample and 43.5% compared to the sample modified with PU400. Further increasing the nanofiller content resulted in a decrease in the value of strain at break, which for C-10-3 was lower than the unfilled P3HB.

The average impact strength (IS) values are shown in Figure 4c. It can be seen that all nanobiocomposites modified with only PU400 or PU400 and 1% Cloisite^®^30B showed higher impact strength than the original sample. However, the maximum impact strength (an improvement of 15% compared to the unfilled P3HB) was shown by the composite designated as C-10-1. It should be noted that the impact strengths of the nanobiocomposites C-10-2 and C-10-3 were lower compared to the unmodified P3HB. As illustrated in Figure 4d, a decrease in the hardness of the obtained nanobiocomposites is evident. The introduction of PU400 made the blend more flexible, and the greatest decrease in hardness was observed from 70 to 42°, i.e., by 15% of such modified P3HB (C-10-0). The addition of organomodified nanoclay increased the hardness of the nanobiocomposite to 64° in reference to C-10-0, but the decreases in the hardness values are still noticeable compared to the unfilled P3HB.

### 3.6. Thermal Stability of the Obtained Nanobiocomposites

Thermogravimetric analysis (TGA) was carried out in order to study the thermal stability of the prepared nanobiocomposites (Table 2). The TGA of the composites showed that the thermal stability of the P3HB increased by 10 °C after the introduction of PU400. Moreover, the thermal stability of the nanobiocomposite containing 1% by weight of Cloisite^®^30B increased by 30 °C compared to the unfilled P3HB. However, a further increase in the amount of nanofiller made the thermal stability of the nanobiocomposite unchanged.

The addition of 3 wt.% of Cloisite^®^30B caused a slight decrease in the thermal stability of the nanobiocomposites, but the thermal stability nevertheless remained superior to that of polyester and polyester modified with 10 wt.% of PU400 (biocomposite designated C-10-0). The fastest decomposition temperatures for P3HB and its hybrid nanobiocomposites were similar (i.e., 282 °C). The exception was P3HB modified with PU400 alone, in which case a temperature drop of 5 °C was observed.

The decomposition residue decreased when PU400 was introduced, and it increased again when Cloisite^®^30B was introduced, and it increased with its increasing content. This was related to the increase in the content of inorganic aluminosilicate in the nanobiocomposite.

### 3.7. Heat Resistance of Nanobiocomposites

Qualitative thermal analysis was carried out based on the heat flow rate of the P3HB and its nanobiocomposites using DSC. Figure 5 shows a comparison of the temperature dependence of the heat flow rate of the unfilled P3HB and the obtained nanobiocomposites. The results indicate that the nanobiocomposites, like P3HB, were semicrystalline nanomaterials with the glass transition and the melting observed during the heating scan. Thermal parameters of the phase transitions were estimated under heating and listed in Table 3. The temperature of the glass transition, T_g_, and the change in the heat capacity at T_g_, ΔC_p_, were established based on the study of the glass transition region, and the heat of fusion, ΔH_f_, and the onset melting temperature, T_m_, were also estimated based on the analysis of the melting region. Additionally, it can be noted that the double melting peak was maintained as with the P3HB analysis. This is probably related to the presence of crystals of different lamella thicknesses [38].

The influence of the Cloisite^®^30B additive on the glass transition temperature (T_g_) and melting temperature (T_m_), and the variations in the heat of fusion and the heat capacity at T_g_ were investigated. Figure 5 shows the curves of the unfilled P3HB and its nanobiocomposites with PU400 and a different content of Cloisite^®^30B. Based on the qualitative analysis (analysis of heat flow rate) of the unfilled semicrystalline P3HB and its nanobiocomposites, the changes in the heat capacity at T_g_ and the heat of fusion with T_m_ were estimated. It was observed that the values of the glass transition temperatures were lower than those of the unfilled P3HB. This is proof of the plasticizing effect of Cloisite^®^30B on the P3HB matrix. The biggest changes in the glass transition temperature of 13.3 °C and the melting temperature of 7.7 °C were observed between the P3HB and C-10-3. The incorporation of polyurethane reduced the glass transition temperature to 0.7 °C [36].

Particularly interesting and useful is the substantial difference between the values of the degradation and the melting temperature, which facilitates processing and prevents the degradation of the material. It can be noted that the best thermal properties and the widest processing window expressed as the difference between the degradation temperature and the melting temperature were obtained for C-10-2 and equaled 95.7 °C (based on T_on_-T_m_ (onset) = 250.0 °C–154.3 °C).

In addition, in order to combine the mechanical properties with the thermal properties, the phase contents of the obtained nanocomposites were analyzed. The changes in the heat of fusion and changes in the heat capacity were analyzed in relation to the values expressed as percentages of the crystalline phase (W_c_) and the mobile (W_a_) and rigid amorphous (W_RAF_) phases (see Table 4).

Figure 6 shows an example analysis of the changes in the heat capacity at the glass transition temperature and the measured heat of fusion for the semicrystalline C-10-1 samples with different thermal histories after different cooling rates. Depending on the different cooling rates, the semicrystalline C-10-1’s glass transition was situated between −4.2 °C and 6.7 °C with a ΔC_p_ (changes in the heat capacity between liquid and solid states at T_g_) in the range 0.11–0.56 J·°C^−1^·g^−1^. The melting endotherm of the semicrystalline C-10-1 occurred between 152 and 161 °C with a heat of fusion in the range 2.7–101.1 J·g^−1^. It should be noted that the changes in the heat capacity values were estimated from the qualitative thermal analysis (analysis of the heat flow rate). The red points are characteristic of a two-phase model. In turn, the black squares confirm the existence of a rigid amorphous fraction and characterize the three-phase model. The solid straight line indicates the full amorphous and full crystalline C-10-1 nanobiocomposite represented by the yellow star points. The change in the heat capacity for the fully amorphous C-10-1 was 0.58 J·°C^−1^·g^−1^, and the heat of fusion for the fully crystalline samples was 114.2 J·g^−1^.

The dependence of the mobile amorphous fraction on the degree of crystallinity of the semi-crystalline C-10-1 is shown in Figure 7. The data obtained from Figure 6 were converted to the corresponding degree of amorphous phase using W_a_ = ΔC_p_/ΔC_p_^100%^ and the degree of crystallinity using W_c_ = ΔH_f_/ΔH_f_^100%^. The degree of crystallinity (W_c_) was defined as the ratio between the experimental heat of fusion (ΔH_f_) of the semicrystalline sample and the heat of fusion of the fully crystalline C-10-1 sample (ΔH_f_^100%^). ΔC_p_ and ΔC_p_^100%^ are the variations in the heat capacity at T_g_ (mobile) of the semi-crystalline and full amorphous C-10-1, respectively. For computation, a ΔC_p_^100%^ value equal to 0.58 J·°C^−1^·g^−1^ and ΔH_f_^100%^ = 114.2 J·g^−1^ were used. The green triangle point derived from the qualitative analysis of the C-10-1 is also plotted in Figure 7. The data of this green triangle point were obtained during heating at 10 °C/min after cooling the sample at a constant rate of 10 °C/min from the melting. It was calculated that the selected C-10-1 sample (green triangle) remained in a three-phase model, which is confirmed in Figure 7 (the sample shows deviation from the straight line). As illustrated in Figure 7, C-10-1 contained 39.8% amorphous and 41.9% crystalline phase, and the rigid amorphous fraction was 18.3% (W_RAF_ = 1 − W_a_ − W_c_). Similar analyses for the other nanocomposites were carried out and are listed in Table 4.

The change in the heat capacity for the fully amorphous material and the heat of fusion for the fully crystalline material were also obtained using the same method as for C-10-1. In Table 4, the phase contents are shown for the samples that were cooled at a controlled rate of 10 °C/min and then heated at a rate of 10 °C/min.

### 3.8. FT IR Analysis for P3HB-Based Hybrid Nanobiocomposites

The FT IR spectra of the P3HB, PU400, and P3HB blends with 10 wt.%. of PU400 and its hybrid nanobiocomposites with Cloisite^®^30B are shown in Figure 8 and Figure 9. The FT IR spectrum of the P3HB shows a characteristic band of ester, i.e., a vibration band of carbonyl groups at 1718 cm^−1^. Moreover, there were bands of asymmetrical and symmetrical C-O bonds of the ester at 1246 and 1129 cm^−1^. Two bands of asymmetrical vibrations and symmetrical C-H bonds of the methyl and methylene groups were observed at 2990 and 2940 cm^−1^. However, no band was observed above 3000 cm^−1^. 

In the FTIR spectrum of PU400 (Figure 8), there was a valence vibration band of the N-H bond of the urethane group in the range of 3600–3100 cm^−1^, which almost disappeared in the spectrum of the C-10-0. A wide band of valence vibrations of asymmetric and symmetric methylene groups of PU400 was observed in the range of 2800–3000 cm^−1^. The band of valence vibrations of the urethane carbonyl groups appeared at 1702 cm^−1^. Bands of the asymmetric and symmetric C-O bonds of the urethane groups are visible at 1271 and 1095 cm^−1^.

In the FTIR spectrum of the hybrid nanobiocomposites (Figure 9), a small band appeared above 3000 cm^−1^, the intensity of which slightly increased with increasing nanoclay contents. In the 2800–3000 cm^−1^ range, there were three bands of asymmetrical and symmetrical C-H bonds of the methyl and methylene groups at 2877, 2933, and 2977 cm^−1^. A combined band of valence vibrations of the C=O ester and urethane groups was observed at 1707 cm^−1^. The common band for esters and urethanes for the vibrations of asymmetric C-O bonds appeared at a wavenumber of 1270 cm^−1^, and the band of symmetric vibrations of C-O bonds appeared at 1097 cm^−1^.

## 4. Discussion

Nanobiocomposites of P3HB modified with Cloisite^®^30B nanoclay and linear aliphatic polyurethane were prepared with the aim of improving the physical properties of the P3HB matrix. Polyurethane (PU400) was obtained by reacting polyethylene glycol with a molecular weight of 400 g/mole with hexamethylene 1,6-diisocyanate (HDI). The introduction of two additives aimed to improve the biopolymer properties resulting from the plasticizing effect of polyurethane, as described earlier [36], and the presence of nanofiller.

Analysis of the diffraction images (Figure 1) of the obtained hybrid nanobiocomposites proved the predominance of the exfoliated structure. The introduction of 1 wt.% of the organomodified nanoclay (Cloisite^®^30B) led to a fully exfoliated structure (C-10-1). On the other hand, the presence of a larger amount of Cloisite^®^30B resulted in possible partial intercalation in addition to delamination (Figure 1), indicating the formation of nanobiocomposites with a disordered, mixed structure.

The TEM images (Figure 2) of the nanobiocomposites show a high degree of nanofiller dispersion, and no influence of the nanoclay concentration on the composite nanostructure was observed. The complete delamination of single silicate nanoplatelets took place in these systems.

The interactions and compatibility of the components of the produced hybrid nanobiocomposites were confirmed by FTIR spectral analysis, indicating the formation of hydrogen bonds between the urethane groups of the modifier and the ester groups of the polyester and the modified montmorillonite (Figure 9).

The SEM images (Figure 3) of the sample fracture surfaces show the mechanism of the interaction of the modifiers by projecting them onto the mechanical properties of the tested nanobiocomposites. SEM analysis of the P3HB modified with PU400 suggested the disentanglement of interacting P3HB chains due to the presence of the PU400, which may have accounted for the increases in the strain at break and impact strength and a slight decrease in the tensile strength of the blend. The significant reduction in hardness was induced by the plasticizing effect of the polyurethane modifier. These results are in contrast to those of the research of Garcia-Quiles et al. who developed P3HB samples loaded with modified montmorillonite (natural sodium montmorillonite modified with a quaternary ammonium salt) at 3 wt.% and evaluated the mechanical properties. The tensile strength values found for a nanocomposite containing 3 wt.% of nanoclay increased by approximately 42% in relation to the nonmodified sample [40]. This may have resulted from the method of composite preparation, as they mixed samples at lower temperatures, but at much higher rotation speeds, which induced higher shear and therefore led to the dispersion and delamination of the montmorillonite.

The addition of 1 wt.%. Cloisite^®^30B indicated the formation of P3HB–PU–MMT adducts that are easily displaced with respect to each other, and this phenomenon may have affected a significant increase in the elongation at break and a slight increase in the impact strength of the tested sample. However, the increase in the hardness of the material may suggest a stiffening effect of the MMT presence.

The addition of quantities greater than 1 wt.%, i.e., 2 wt.% or 3 wt.% of Cloisite^®^30B already excludes the formation of the mentioned adducts inherent in the C-10-1 sample and indicates that the addition of 2% or 3% Cloisite^®^30B is already useless. The structures of these nanobiocomposites were similar to P3HB containing only PU400, and they were more similar to a binary biocomposite with more nanofiller added. Increasing the amount of MMT above 1 wt.% did not affect the material’s elasticity, which appeared to decrease the impact strength, toughness, and relative elongation at break of these nanobiocomposites and suggests the stiffening of the P3HB–PU400 samples by such amounts of Cloisite^®^30B (2 and 3 wt.%).

The obtained P3HB nanobiocomposites showed a predominance of an exfoliated structure, but a decrease in the content of nanofiller resulted in only the exfoliated structure, which had an impact on the mechanical properties. Therefore, the best mechanical properties were shown by the composite designated as C-10-1, containing 1 wt.% of Cloisite^®^30B and 10 wt.% of PU400, whose impact resistance enhancement was due to the exfoliated structure and the formation of the P3HB-PU-MMT adducts as observed on the SEM micrographs. The addition of organomodified nanoclay resulted in a stiffening of the exfoliated structure and an increase in the hardness of the composite but below the hardness value of the unfilled P3HB.

The thermal properties of the produced nanobiocomposites were improved as expected. The addition of the PU400 modifier caused an increase in the onset of the degradation temperature by 10 °C, and a further increase of 20 °C or 30 °C due to the addition of 1 wt.% of Cloisite^®^30B compared to the unfilled P3HB. Increasing the amount of nanofiller not only failed to further increase the thermostability but even induced its decrease. Nanocomposites with Cloisite^®^30B alone were obtained by Naguib using an intercalation method. It was observed that the greater the addition of nanoclay, the greater the reduction in the composite decomposition temperature. It should be noted that 5 and 10 wt.% of Cloisite^®^30B were used, and only a deterioration in the thermal properties of the composites was observed compared to the neat P3HB [41].

The addition of polyurethane caused the plasticization of P3HB, which appeared in the form of a decrease in the glass transition temperature, while the addition of nanoclay caused a stiffening effect and a further increase in the glass transition temperature as the nanoclay content increased. In summary, the plasticizing effect was not completely suppressed by the presence of nanofiller because the value of the glass transition temperature of the nanocomposites remained lower than the glass transition temperature of the unfilled P3HB. The tendency of the Tg to decrease has been observed in other studies [40,42,43]. Furthermore, the addition of Cloisite^®^30B led to a decrease in the degree of crystallinity of nanobiocomposites in reference to P3HB. The X-ray analysis of the prepared nanobiocomposites also confirmed an increase in the amorphousness of the nanobiocomposites compared to the semi-crystalline unfilled P3HB.

From the comparison of the melting point values of the individual nanobiocomposites and their degradation onset temperatures (Table 2 and Table 3), it was possible to notice a definite increase in their differences, which indicates an increase in the processing window of the produced P3HB hybrid nanobiocomposites compared to the unfilled P3HB up to almost 100 °C.

This increases the possibilities of using P3HB for the production of disposable packaging materials or in medical applications, e.g., as implants, drug carriers, etc. Preliminary studies of nanofiber formation and active substance release involving P3HB and cytisine have already been carried out [44].

## 5. Conclusions

The conducted tests clearly showed the influence of the dispersion degree of Cloisite^®^30B nanoclay in the P3HB matrix with the addition of aliphatic linear polyurethane (PU400) on selected thermal and mechanical properties of the produced nanobiocomposites.

The polymer matrix chains penetrated the interlayer spaces of Cloisite^®^30B, and delamination occurred with partial intercalation as the proportion of nanoclay in the matrix increased. This was possible due to the strong interaction between the components and the shear forces during direct mixing in a co-rotating twin-screw microextruder.

The morphology studies on the produced nanobiocomposites showed the compatibility of the components used and the homogeneity of the obtained structure, the plasticizing effect of PU400, and the stiffening effect of Cloisite^®^30B. The polyester matrix modified with 10% PU400 and the smallest amount of Cloisite^®^30B (1 wt.%) was characterized by the formation of P3HB-PU-MMT adducts easily displaced with respect to each other, which positively affected the mechanical properties of the nanobiocomposites, i.e., better impact strength and lower hardness. The best mechanical properties were shown by a nanocomposite containing 1 wt.%. Cloisite^®^30B (15% increase in impact strength, 60% increase in relative elongation at break, and 15% decrease in hardness).

The introduction of nanoplatelets into the polymer matrix resulted in better thermal stability of the nanocomposites as measured by the TGA method under non-oxidizing conditions. There was an increase in the degradation temperature by 30 °C. As the concentration of the nanoplatelets increased, the thermal stability of the nanocomposites deteriorated but remained better compared to the unfilled P3HB.

Qualitative thermal analysis showed a decrease in the glass transition and melting temperature of the P3HB nanobiocomposites. A reduction of the melting temperature was beneficial due to easier processing and prevented material degradation.

The obtained nanobiocomposites containing the smallest amount of nanofiller (1 wt.%) exhibited an exfoliated structure and formed P3HB-PU-MMT adducts, which were characterized by the best mechanical and thermal properties.

## Data Availability

Not applicable.

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
