# Peer review of "Polymer/Layered Clay/Polyurethane Nanocomposites: P3HB Hybrid Nanobiocomposites—Preparation and Properties Evaluation"

_nanomaterials, 2023, doi:10.3390/nano13020225_

Round 1
Reviewer 1 Report
The manuscript introduces a kind of hybride nanobiocomposites with P3HB matrix filled by PU400 and nanoclay and investigates their nanostructure, mechanical and thermo-physical properties. The results should be significant for engineering uses of the nanocomposites. The reviewer provides following revisions and suggestions as references to the editor and authors:
(1) For the microstructure of the nanocomposites as shown in Fig. 2 and Fig. 3, some typical structural characteristics should be marked out, e. g. P3HB chains, silicate gallery, as well as the exfoliated structure and P3HB-PU-MMT adducts hereafter mentioned in line 514-516, to facilitate the readers. In addition, the description “therefore favouring dispersion and delamination of montmorillonite” is short of evidence without the apparent identifications in the microstructural images.
(2) TG and DTG curves do not appear in Table 2. Only some characteristic values of the curves are shown in Table 2.
(3) The definitions should be unified through the context, e. g. P3HB-2 in line 372 and C-10 in line 441 seem to be C-10-2 and C-10-0, respectively? In addition, how does the value 95.7°C deduced from in line 372?
(4) Degradation temperature should be defined and its determination should be described according to the TG, DTG, and DSC curves.
(5) As for Fig. 9, line 448-449, the description “a small band appears above 3000 cm-1, the intensity of which increases with increasing nanoclay content” seems to be inconsistent with the curves in Fig. 8 and Fig. 9.
(6) There are some inappropriate descriptions to be revised as follows: Line 273-274, “different structure than” should be “different structure from”; Line 292, “the value of stress at break” should be “the value of strain at break”.
Author Response
Thank you very much for so kind review. All changes and correction were marked in red.
(1) For the microstructure of the nanocomposites as shown in Fig. 2 and Fig. 3, some typical structural characteristics should be marked out, e. g. P3HB chains, silicate gallery, as well as the exfoliated structure and P3HB-PU-MMT adducts hereafter mentioned in line 514-516, to facilitate the readers. In addition, the description “therefore favouring dispersion and delamination of montmorillonite” is short of evidence without the apparent identifications in the microstructural images.
Ad.1. Thank you for the comment. The typical structural characteristics in Fig. 2 and Fig. 3 were marked out. Usually, they are not marked out in the research papers but you are right that it is facility for readers.
(2) TG and DTG curves do not appear in Table 2. Only some characteristic values of the curves are shown in Table 2.
Ad. 2. Thanks for your remark. The Table 2 description was improved as follows: The selected characteristic values of the TG and DTG curves of P3HB and nanobiocomposites based on P3HB, polyurethane (PU400) and Cloisite®30B recorded at heating rate of 5°C/min in nitrogen atmosphere (designated as C-10-1, C-10-2 and C-10-3, respectively).
(3) The definitions should be unified through the context, e. g. P3HB-2 in line 372 and C-10 in line 441 seem to be C-10-2 and C-10-0, respectively? In addition, how does the value 95.7°C deduced from in line 372?
Ad. 3. The named of nanocomposites were unified and the value of 95.7°C linking with widest processing window was explained for C-10-2 as follows: It can be noted that the best thermal properties and the widest processing window expressed as the difference between the degradation temperature and the melting temperature were obtained for C-10-2 and equals 95.7°C (based on Ton - Tm(onset) = 250.0°C - 154.3°C).
(4) Degradation temperature should be defined and its determination should be described according to the TG, DTG, and DSC curves.
Ad. 4. The temperature of degradation values were established based on the onset of decomposition (Ton) using the TG curves by thermogravimetric analysis (Metler Toledo TGA/DSC 3+ thermogravimetric analyzer). Moreover, the other thermal parameters e.g. temperature of glass transition, change of heat capacity at Tg, melting temperature and heat of fusion were estimated based on the qualitative thermal analysis of heat flow rate for each nanocomposites based on the DSC measurements (Q1000TM from TA Instruments, Inc. (New Castle, DE, USA)). The degradation of temperature hasn't observed for DSC analysis because this calorimeter is the heat-flux type and measurements are harmful for the measuring cell. The thermogravimeter is suitable for determining the degradation temperature. In addition, our thermogravimeter has the ability to view a DSC scan, but these measurements are less accurate than in case of the DSC Q1000 measurements, which is dedicated to the qualitative and quantitative thermal analysis of advanced materials.
(5) As for Fig. 9, line 448-449, the description “a small band appears above 3000 cm-1, the intensity of which increases with increasing nanoclay content” seems to be inconsistent with the curves in Fig. 8 and Fig. 9.
Ad. 5. The changes are very discreet, so the text has been changed to “a small band appears above 3000 cm-1, the intensity of which slightly increases with increasing nanoclay content”
(6) There are some inappropriate descriptions to be revised as follows: Line 273-274, “different structure than” should be “different structure from”; Line 292, “the value of stress at break” should be “the value of strain at break”.
Ad. 6. Thank you for corrections. They were done in the main text.
Reviewer 2 Report
This is a well-written paper, clearly presented. My detailed comments can be found in the attached file

Author Response
Thank you very much for so kind review. All changes and correction were marked in red.
The Abstract and Methods sections do not mention the use of IR spectroscopy.
Ad. The omitted information about IR spectroscopy was added.
Line 86: refers to” SEC in THF”. Readers who are not organic chemists will benefit from a clearer explanation of these abbreviations.
Ad. You’re right. The abbreviations were explained in the manuscript text.
In contrast, the SEM is not named, though the brand name is given for the sputter coater used to prepare specimens.
Ad. The SEM was named in the manuscript text.
Likewise the XRD is not identified. The insturment used for DSC is identified, but not the instrument used for TGA.
Ad. The names of XRD and TGA equipment were identified in the text.
As a very minor point, the XRD scanning conditions are listed as being carried out from 0-30 degrees over 3 hours. It would perhaps be more useful to describe a scan rate (e.g., degrees per minute at a specified step increment). I presume the analysis was done by step scanning; it’s been many years since I’ve seen anyone use an analog (continuous motion) for an XRD goniometer. Also, I don’t think it is possible to have 0 degrees as the starting angle. It would mean that the X-ray tube would be aimed directly at the radiation detector, and most systems won’t allow that geometry. Later, the text states that the starting angle was 1 degree (line 208).
Ad. Thank you for the comment. All inaccuracies have been corrected in the description of the method and given in the text of the article.
Lines 176, 206: These are among several places where the font size suddenly change.
Ad. The font size was checked. Whole text has unified font size. This misunderstanding results from conversion word file to pdf file.
Line 227: Needs a comma after “Moreover”
Ad. It was corrected.
Figure 2. The most prominent visual features of these images are the swirls that I presume are artifacts of preparation. They need to be mentioned in the caption.
Ad. The comment was added in the figure caption.
Line 250. Should be 250X magnification. But this is deceptive because the magnification listing only applies to the size of the image on the SEM viewing screen. This will be different if the image is reproduced at any other size. The meaningful magnification indication is the scale bar.
Ad. The magnification mark was corrected. The image was not reproduced at any other size.
Line 263. Delete comma after (PU400).
Ad. It was done.
Line 273 mentions that Cloisite clay is montmorillonite. If this is important information, it should be presented when Cloisite is first mentioned.
Ad. It was corrected in the manuscript text.
Line 303: Should be “impact strengths..... are lower” . The description refers to properties of two materials.
Ad. It was corrected.
Table 4.
Abbreviations in the table heading line should be explained.
Ad. Abbreviations were explained below the Table 4.
